# Is Supine Position Superior to Prone Position in the Surgical Pinning of Supracondylar Humerus Fracture in Children?

**DOI:** 10.3390/jfmk5030057

**Published:** 2020-07-31

**Authors:** Vito Pavone, Andrea Vescio, Maria Riccioli, Annalisa Culmone, Pierluigi Cosentino, Marco Caponnetto, Sara Dimartino, Gianluca Testa

**Affiliations:** 1Department of General Surgery and Medical Surgical Specialties, Section of Orthopaedics and Traumatology, Surgery, AOU Policlinico-Vittorio Emanuele, University of Catania, 95123 Catania, Italy; annalisa.culmone@libero.it (A.C.); pierluigi-cosentino@hotmail.it (P.C.); caponnettomarco@tiscali.it (M.C.); saradimartino1@gmail.com (S.D.); gianpavel@hotmail.com (G.T.); 2Dipartimento Area Chirurgica, U.O.C. di Ortopedia e Traumatologia, Ospedale Umberto I di Siracusa, 96100 Siracusa, Italy; marywe@hotmail.it

**Keywords:** pediatric fractures, supracondylar humerus fracture, pinning techniques, supine position, prone position

## Abstract

Background: Supracondylar humerus fracture (SCHF) is a frequent injury in pediatric ages. Closed reduction and percutaneous pin fixation is a common treatment of displaced SCHF. Surgery is usually performed in the supine position; otherwise the prone position allows an easier fracture reduction and a safe placement of pins. The aim of study is to compare the clinical and radiographic results of the treatment of displaced SCHF, comparing two different intra-operative positionings. Methods: 59 SCHF affected children were retrospectively divided into supine (Group 1; *n* = 34) and prone (Group 2; *n* = 25), according to intraoperative position. All treated subjects were clinically evaluated according to Flynn’s criteria and Mayo Elbow Performance Score, and radiographically, including the measurement of the Baumann angle. Results: Clinically, Group 1, according Flynn’s criteria, had excellent cosmetic outcome in 32 subjects (94.1%). Mean MAYO Score was 96.0 ± 3.8. Group 2, according Flynn’s criteria, had excellent cosmetic outcomes in 23 subjects (92.0%). Mean MAYO Score was 97.8 ± 3.3. Radiographically, mean difference of Baumann’s angle between the injured limb and the normal limb was 5.5° ± 1.0° in Group 1 and 5.1° ± 1.1° in Group 2. Conclusion: Both supine and prone positioning achieved a satisfying outcome with similar results in joint function recovery and complications.

## 1. Introduction

Supracondylar humerus fractures (SCHF) represent 3% of all pediatric fracture, the most common elbow fracture [1,2,3] and the second most frequent of upper limb after distal radius fractures [2,3]. Male gender was more frequent, but, after the increase in sports activities the sex ratio is similar. Accounting 90–98% of injuries, the extension trauma is the most frequent pathogenesis mechanism [4]. The gold standard technique is closed reduction and percutaneous pinning (CRPP) [5,6]. Acceptable rotation is achieved if the medial and lateral columns are well aligned [1]. CRPP is a mini-invasive procedure and a safe technique with low rate of complications (2–8%) [7]. Open reduction is uncommon and limited to selected fractures, such as irreducible fractures by interposition of soft tissues or in cases of neurovascular complications [8,9]. Generally, the supine position is the ordinary placement during surgery [10], but recent findings have shown prone position as an efficient alternative. Despite a more challenged airway management [11], prone position achieves a fracture reduction more easily to perform and, at the same time, a safer pins placement can be performed avoiding excess of elbow flexion [12,13].

The purpose of the study is to analyze and compare the clinical and radiological outcomes in SCHF affected patients treated with CRPP in supine or prone position.

## 2. Materials and Methods

### 2.1. Sample

A retrospective medical record review (1st May 2005 to 31st July 2014) was performed, including children younger than 12 years of age who underwent CRPP for displaced SCHFs at our institution. The study was conducted according to the declaration of Helsinki.

The inclusion criteria were as follows: (1) confirmed diagnosis of Gartland type III; (2) patient age under 12 years; (3) trauma in extension resulting from a fall on the palm of the hand with a hyperextended elbow; (4) CPRR treatment within 8 h according the “treatment protocol” described below; (5) at least 6 months of follow-up; (6) complete clinical and radiographic data.

The exclusion criteria were: (1) poly-traumatized patients with other associated fractures; (2) open or pathological fracture and (3) No complete clinical or radiographic data. According to inclusion and exclusion criteria, 59 subjects were eligible for the study. Informed consent was obtained by all patients.

All patients were admitted through the emergency department with the following demographic and clinical data captured: gender, age at the time of trauma, mechanism of accident, involved side, presence or absence of associated neurovascular injury, and whether the fracture was closed or open.” In addition, the intraoperatively patient position was obtained from the medical records (Table 1).

Intra-operative positioning choice depended upon surgeon experience and fracture pattern (Figure 1).

Based on the patient position during the surgery, the samples were divided into groups: supine (Group 1; *n* = 34) and prone (Group 2; *n* = 25) (Table 1). The cohorts were found similar per demographics characteristics (*p* > 0.05)

### 2.2. Treatment Protocol

All subjects were treated within eight hours from presentation with a closed reduction and percutaneous pin fixation. After inducing general anesthesia, the arm was positioned at a “C” on the brightness amplifier plate. Intra-operative positioning could be supine or prone. According to the age of patients, closed reduction and fixation with 1.6 to 2 mm K-wire were used. The adopted configurations were cross-pins or lateral pins [3].

After stabilization with wires, the children were immobilized with a simple posterior splint at 90 degrees of flexion. A clinical and radiographic evaluation was performed at 7 days from surgery. The wires were removed after 4 weeks, to start a rehabilitation protocol to restore the full range of motion of the elbow. All patients were assessed at 1, 3, 6, and 12 months and then every year.

### 2.3. Clinical and Radiological Assessments

All treated subjects were evaluated clinically according to Flynn’s criteria [14] (Table 2) and the Mayo Elbow Performance Score (MEPS) (0–100 points) (Table 3) [15,16] and range of motion (ROM) flexion-extension, as well as, supination and pronation. Both cohorts underwent X-rays at their last follow-up in anteroposterior view and their medicated Baumann’s angles [17] were measured.

### 2.4. Statistical Analysis

Continuous data are presented as means and standard deviations, as appropriate. The independent sample Student’s *t*-test and Fisher’s exact test were used to compare age, follow-up, MEPS, range of motion for flexion-extension, supination-pronation, and modification of Baumann’s Angle means between the groups. The χ^2^-test was used to verify the homogeneity of the three groups based on gender and laterality—Flynn’s criteria. The selected threshold for statistical significance was *p* < 0.05. All statistical analyses were performed using the 2016 GraphPad Software (GraphPad Inc., San Diego, CA, USA). 

## 3. Results

### 3.1. Sample

Group 1 was composed of 34 patients, 22 (64.7%) males and 12 (35.3%) females. The right side was involved in 18 (52.9%) cases, and the left side in 16 (47.1%). Mean age at time of surgery was 6.1 ± 2.8 (range 1.0–12.8), mean follow-up was 59.8 ± 15.8 months (range 24–79). Group 2 was composed of 25 patients, 12 (48%) male and 13 (52%) female. The right side was involved in 14 (56%) cases, the left side in 11 (44%). Mean age at time of surgery was 5.9 ± 2.3 (range 1.5–10.5). Mean follow-up was 59.9 ± 12.8 months (range 27–78) (Table 1).

### 3.2. Clinical and Radiological Assessments

Group 1, according Flynn’s criteria, had an excellent cosmetic outcome in 32 subjects (94.1%) and good in 2 (5.9%). The functional factor was satisfactory in 33 (97%) patients. Mean MEPS was 96.0 ± 3.8 (range 86–100). On examination at final follow-up, the range of motion in the treated arm resulted in values for flexion of 110.9° ± 14.3° (range 90°–135°), extension of 3.4° ± 1.8° (range 0°–10°), and supination to pronation of 84.9° ± 3.1° (range 81°–90°). Group 2, according Flynn’s criteria, had excellent cosmetic outcome in 23 subjects (92.0%) and good in 2 (8.0%). Functional factor was satisfactory in 100% of patients. Mean MEPS was 97.8 ± 3.3 (range 91–100). At the final follow-up, the range of motion gave a flexion range of about 113.6° ± 11.2° (range 94°–137°), extension of 2.9° ± 2.2° (range 0°–10°), and supination to pronation of about 86.2° ± 2.2° (range 84°–90°). 

Radiographically, mean difference of Baumann’s angle between the injured limb and the normal limb was 5.5° ± 1.0° (range 2.7°–6.7°) in Group 1 and 5.1° ± 1.1° (range 2.8°–6.6°) in Group 2. Minor complications encountered. Group 1: asymmetry of 0.8 cm in one (2.9%) case, ulnar nerve paresthesia in 2 cases (5.9%), which resolved spontaneously in about 2–3 months. Six degrees of varus deviation and 7.5° deficit of extension were reported in 1 case (2.9%). Group 2: mild hyperextension in one case (4%), local infection treated with antibiotic therapy in two cases (8%), spontaneous removal of K-wires in one case (4%). No other major complications were reported.

Daily activities and sport participation were restored for all patients. Statistical analysis showed no significant differences between groups (Table 4).

## 4. Discussion

According our data, supine and prone intraoperative positions are similar for functional and radiological outcomes. At the same time, no differences between both approaches were found, considering complication rate and anesthesia management. SCHFs were widely investigated in literature, but the management and the treatment are still debated; there is no common consensus among the pediatric orthopedics. Despite several societies and expert research committees [18,19,20,21,22] being involved in the development of guidelines and management algorithms, many recommendations are poorly supported by the literature evidence. 

Timing of treatment, pin configuration and intraoperative patient position are the most controversial issues. In fact, to postpone surgery could significantly influence the necessity of open reduction approach, complication, or overall outcome [23]. At the same time, several methods have been described. The cross-wired and lateral wires configurations are the most examined [3,24,25,26,27,28]: cross-wired is more biomechanically stable [26,27], lateral reduces nerve injury risk [25]. 

In 2018, the Italian Pediatric Orthopedic and Traumatology Society (SITOP) guidelines did not highlight the superiority of supine or prone position, considering radiation exposure, surgery length, reduction attempts, clinical and radiological outcome [29,30], and leaving positioning choice to surgeon experience. 

Despite the SITOP recommendations, a multicenter study of the main Italian national pediatric trauma centers noted that only 53 out of 529 (10%) patients were surgically treated in prone position [22]. In our series, more than 40% of the sample was treated in prone position; these patients reported excellent aesthetic and functional outcomes, according to Flynn’s Criteria. However, supine positioning patients recorded 94% of excellent results. De Pellegrin et al. [5] highlighted 100% of excellent functional outcomes in patients treated in prone position. K. Venkatadass et al. [30] reported the 87.5% and 89.1% of satisfaction results for prone and supine position, respectively. Guler et al. [13] compared 27 prone positioned patients to 29 supine and did not found any differences. Hsuan-Kai Kao et al. [31] treated in prone position 10 patients in 7 years, and 90% of them had excellent outcomes, according to Flynn’s criteria. 

In our study, as in the series of Gurler et al. [13], both intraoperative positionings reported good and comparable (*p* = 0.15) radiographic outcomes, according to modified Baumann’s angle. Hsuan-Kai Kao’s study [31] showed that the mean change in Baumann’s angle was 3.5.

Considering the equal clinical and radiological results, different authors have listed the advantages of prone intraoperative positioning, such as a simpler reduction, a more comfortable use of C-arm and a reduction in nerve injuries. In case of posterior displacement, the gravity force correctly positions the distal humerus in the frontal plane [32]. Moreover, an easier gravity-guided reduction avoids the elbow hyperflexion, generally needed in patients treated in supine position [33]. During elbow hyperflexion, the ulnar nerve tends to slip anteriorly, out of the cubital tunnel, passing over the medial epicondyle; nerve hypermobility increases iatrogenic injuries through the medial pinning [33]. In our sample, minor complications were encountered, but no nerve injuries; equally, Havlas et al. [11] did not document ulnar nerve iatrogenic lesions in 455 SCHFs-affected children treated in prone position. On the other hand, the supine patient position requires minimal time [11], allowing for standard anesthesia management and the possibility to use an anterior, antero-medial or antero-lateral approach, if open reduction is needed [32,34,35]. 

However, as well as the patient position, other parameters can influence the outcome. The fixation with elastic stable intramedullary nail (ESIN) after close reduction was described [36,37,38]. ESIN advantages include cast-free treatment after surgery and the protection of the ulnar nerve by introducing the ESIN at the proximal humerus. When compared to K-wire fixation, ESIN are rarely suggested, because they cannot be applied for more complex fractures, due to the necessity of anatomical reduction, to avoid cubitus varus [39,40].

Surgeon experience is another influencing factor for SCHFs-treated patient outcome [41]; Tuomilehto et al. [42] reported a slightly higher rate of complications, but no differences in the quality of reduction between residents and senior orthopaedic surgeons; otherwise, the open reduction rate increases when the surgeon is a resident [8]. 

The limitationss of the study are its retrospective nature, small size, and the lack of objective measurements.

## 5. Conclusions

In conclusion, regarding the surgical pinning of SCHFs, the supine intraoperative positioning is not superior to prone. In both groups, satisfying outcomes were achieved, with similar results in joint function recovery and complications. Surgeon experience is crucial in positioning choice.

## Figures and Tables

**Figure 1 jfmk-05-00057-f001:**
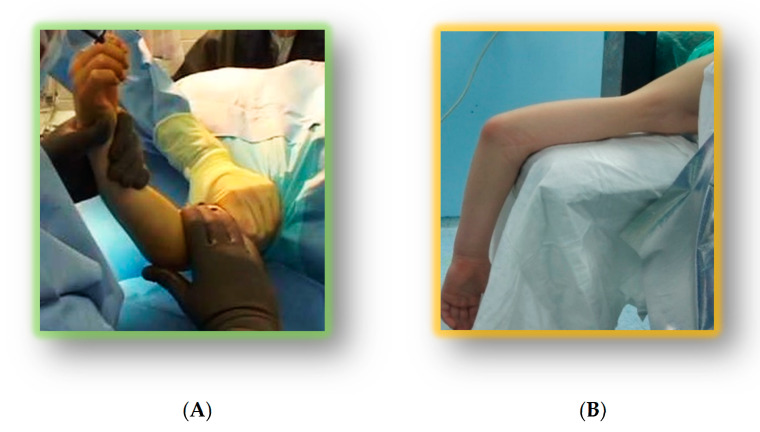
(**A**) Supine positioning; (**B**) Prone positioning.

**Table 1 jfmk-05-00057-t001:** Groups demographics characteristics.

Characteristics	Group 1—Supine Position (*n* = 34)	Group 2—Prone Position (*n* = 25)	*p*-Value
Age (years)	6.1 ± 2.8	5.9 ± 2.3	0.77
Gender (Male/Female)	22/12 (64.7%)	12/13 (48%)	0.66
Left Side (Left/Right)	18/16	14/11	0.53
Follow up (months)	59.8 ± 15.8	59.9 ± 12.8	0.74

**Table 2 jfmk-05-00057-t002:** Flynn criteria.

Result	Rating	Cosmetic Factor Loss of Carrying Angle (degrees)	Functional Factor Motion Loss (degrees)
Satisfactory	Excellent	0–5	0–5
	Good	5–10	5–10
	Fair	10–15	10–15
Unsatisfactory	Poor	>15	>15

**Table 3 jfmk-05-00057-t003:** Mayo Elbow Performance Score.

Feature	Rating	Score
Pain Intensity	None	45
	Mild	30
	Moderate	15
	Severe	0
Arc of Motion	>100°	20
	50°–100°	15
	<50°	5
Stability	Stable	10
	Moderate instability	5
	Moderate instability	0
Function	Can comb hair	5
	Can eat	5
	Can perform hygiene	5
	Can put on shirt	5
	Can lace shoe	5
Maximum Score		100

**Table 4 jfmk-05-00057-t004:** Clinical and Radiological assessment results.

		Group 1 (*n* = 34)	Group 2 (*n* = 25)	*p*-Value
	Excellent	32 (94.1%)	25 (100%)	0.22
Flynn Criteria	Good	2 (5.9%)	0
	Fair	0	0
	Poor	0	0
Extension	3.4 ± 1.8	2.9 ± 2.2	0.35
Flexion	110.9 ± 14.3	113.6 ± 11.2	0.44
Supination-Pronation	84.9 ± 3.1	86.2 ± 2.2	0.08
Mayo Elbow Performance Score	96.0 ± 3.8	97.8 ± 3.3	0.06
Modified Baumann’s Angle (deg)	5.5 ± 1.0	5.1 ± 1.1	0.15
Reoperations	0	0	
Minor Complications	4 (11.4%)	3 (12%)

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
