# Peer review of "Is Supine Position Superior to Prone Position in the Surgical Pinning of Supracondylar Humerus Fracture in Children?"

_jfmk, 2020, doi:10.3390/jfmk5030057_

Round 1

Reviewer 1 Report

EVERYTHING IS WELL DONE.

The paper titled "Is supine position superior to prone position in 3 surgical pinning of supracondylar humerus fracture 4 in children?" is a very well done article: the Background in good argued, Materials and Methods are well described, despite of non large number of patients analyzed, all the variables and features are well described and represented in the all reported tables.
References are not too dated and recent. In my opinion is a good work.

Author Response

Thank you for your revision and evaluable comments.

Reviewer 2 Report

Valuable article for clinicians with worthy information.

Author Response

Thank you for your revision e your evaluable comment.

Reviewer 3 Report

The authors present results of a so called retrospective review of clinical data comparing prone and supine positioning of patients during percutaneous pin fixation for supracondylar fractures. The manuscript needs extensive editing for english language and improvement of scientific presentation of results. Furthermore, follow-up examinations at certain time points as well as the acquisition of x-ray images of the contralateral unaffected limb (there is no need of radiation exposure of the contralateral limb, even in pediatric patients) indicate that this was a prospective study rather than a retrospective review. If there was no ethics approval of the study beforehand, this manuscript cannot be published.

Line 51-54: please define inclusion and exclusiuon criteria first, then describe which parameters you have evaluated in your study

Line 57-59: Parameters 4) and 6) are essentially the same?

Line 60-61: if inclusion is <12 years, it is not necessary to have an exclusion criterion > 12 years

Line 62ff and 70ff: what was the reason at time of operation to put the patient in a supine or prone position, personal preference of the operating surgeon?

Line 68ff: example images should be shown of the percutaneous pin fixation in prone and supine position

Line 83-84: This is follow-up protocol is extensive and not common clinical practice, the study was hopefully an ethics approved prospective study rather than a retrospective review. At which timepoints did you acquire x-ray images. Acquisition of x-ray images of the normal limb for comparison is not clinically indicated (Line 116ff). Please state the nature of the study clearly and provide ethics approval.

Line 110ff: the range of motion should be documented according to common standards, i.e. range of motion for the elbow 10-0-150 according to the “Neutralnullmethode”… (flexion-extension of 107.5°±15.3 alone does not provide meaningful information) For statistical analysis it might be easier to document extension and flexion separately including negative values for an extension/flexion deficit.

Line 129ff: The discussion section needs extensive editing of the English language.

Author Response

Thank your for revising our manuscript. Here you will find your suggestions (Q) and our answers (A).

The authors present results of a so called retrospective review of clinical data comparing prone and supine positioning of patients during percutaneous pin fixation for supracondylar fractures.

Q1) The manuscript needs extensive editing for english language and improvement of scientific presentation of results.

A1) An extensive editing from a native English editor has been performed.

Q2) Furthermore, follow-up examinations at certain time points as well as the acquisition of x-ray images of the contralateral unaffected limb (there is no need of radiation exposure of the contralateral limb, even in pediatric patients) indicate that this was a prospective study rather than a retrospective review. If there was no ethics approval of the study beforehand, this manuscript cannot be published.

A2) We confirm that this is a retrospective study. For a strict Supracondylar humeral fractures follow-up, to prevent complications, we perform comparative elbows x-ray.

Q3) Line 51-54: please define inclusion and exclusiuon criteria first, then describe which parameters you have evaluated in your study

A3) According to your suggestion, we modified the paragraphs.

Q4) Line 57-59: Parameters 4) and 6) are essentially the same?

A4) We agree. Parameter 6 was deleted.

Q5) Line 60-61: if inclusion is <12 years, it is not necessary to have an exclusion criterion > 12 years

A5) Your suggestion was followed.

Q6) Line 62ff and 70ff: what was the reason at time of operation to put the patient in a supine or prone position, personal preference of the operating surgeon?

A6) The reason depends on surgeon experience and fracture pattern.

Q7) Line 68ff: example images should be shown of the percutaneous pin fixation in prone and supine position

A7) Required images have been added.

Q8) Line 83-84: This is follow-up protocol is extensive and not common clinical practice, the study was hopefully an ethics approved prospective study rather than a retrospective review. At which timepoints did you acquire x-ray images. Acquisition of x-ray images of the normal limb for comparison is not clinically indicated (Line 116ff). Please state the nature of the study clearly and provide ethics approval.

A8) Sorry, but we confirm that this is a retrospective study. For a strict SHF follow-up, to prevent complications, we perform comparative elbows x-ray.

Q9) Line 110ff: the range of motion should be documented according to common standards, i.e. range of motion for the elbow 10-0-150 according to the “Neutralnullmethode”… (flexion-extension of 107.5°±15.3 alone does not provide meaningful information) For statistical analysis it might be easier to document extension and flexion separately including negative values for an extension/flexion deficit.

A9) Following your suggestions, the phrases have been modified.

Q10) Line 129ff: The discussion section needs extensive editing of the English language.

A10) The phrases were modified.

Round 2

Reviewer 3 Report

The paper has been edited and is now improved, but I still have concerns regarding the retrospective nature of the study and/or the clinical practice at this institution in terms of performing x-rays of uninjured, healthy limbs in children.

Line 58: informed consent was obtained for surgery or for being part of this study? When was the infomed consent obtained? Retrospectively or before inclusion in the study?

Line 76: please do not divide the table into two halfs!

Line 90ff: I do not understand how you define „last follow-up“ with x-rays and have a broad range of follow-ups between months and years. If this is only a retrospective review of medical records. In clinical practice, how did you define a visit of the patient in the outpatient clinic as last follow-up?

Performing x-rays of the uninjured, contralateral side for comparison is not justified in clinical routine and may only be justified in a prospective clinical trial if parents have given their informed consent to a sole research x-ray exam of an uninjured limb. If you routinely perform x-rays of uninjured limbs for comparison please revise your clinical practice and get in touch with the respective Italian and European Societies.

Line 128ff: Please do not divide the table

Author Response

Q1) Line 58: informed consent was obtained for surgery or for being part of this study? When was the informed consent obtained? Retrospectively or before inclusion in the study?

A1) Thank for the comment. As reported in Methods, the patients had at least 6 months of follow-up. The informed consent was obtained before the inclusion in the study, during a follow-up

Q2) Line 76: please do not divide the table into two halfs!

A2) Thank for the comment. The requested modifies were made

Q3) Line 90ff: I do not understand how you define „last follow-up“ with x-rays and have a broad range of follow-ups between months and years. If this is only a retrospective review of medical records. In clinical practice, how did you define a visit of the patient in the outpatient clinic as last follow-up?

A3) Thank for the comment. Routinely we performed clinical follow-up 1 week, 1, 3, 6 and 12 months after the treatment and then every year until the skeletal maturation.

Q4) Performing x-rays of the uninjured, contralateral side for comparison is not justified in clinical routine and may only be justified in a prospective clinical trial if parents have given their informed consent to a sole research x-ray exam of an uninjured limb. If you routinely perform x-rays of uninjured limbs for comparison please revise your clinical practice and get in touch with the respective Italian and European Societies.

A4) At 1 week, 1, 12 and 24 follow-up the patients underwent to radiological exam.  In one- and two-years radiological follow-up, after the obtain of parent informed consent, a comparison between the injured and uninjured limb is performed to rule out varus or valgus deformities of elbow.

Q5) Line 128ff: Please do not divide the table

A5) Thank for the comment. The requested modifies were made